# Clinical Applications of Cell-Scaffold Constructs for Bone Regeneration Therapy

**DOI:** 10.3390/cells10102687

**Published:** 2021-10-08

**Authors:** Venkata Suresh Venkataiah, Yoshio Yahata, Akira Kitagawa, Masahiko Inagaki, Yusuke Kakiuchi, Masato Nakano, Shigeto Suzuki, Keisuke Handa, Masahiro Saito

**Affiliations:** 1Department of Restorative Dentistry, Division of Operative Dentistry, Graduate School of Dentistry, Tohoku University, Sendai 980-8575, Japan; yahataendo@tohoku.ac.jp (Y.Y.); kitagawa@osterenatos.jp (A.K.); y.kaki@tohoku.ac.jp (Y.K.); masato.nakano.q4@dc.tohoku.ac.jp (M.N.); shigeto.suzuki.p5@dc.tohoku.ac.jp (S.S.); handa@kdu.ac.jp (K.H.); mssaito@dent.tohoku.ac.jp (M.S.); 2OsteRenatos Ltd., Sendai Capital Tower 2F, 4-10-3 Central, Aoba-ku, Sendai 980-0021, Japan; 3National Institute of Advanced Industrial Science and Technology, 2266-98 Anagahora, Nagoya 463-8560, Japan; m-inagaki@aist.go.jp; 4Department of Oral Science, Division of Oral Biochemistry, Graduate School of Dentistry, Kanagawa Dental University, Yokosuka 238-8580, Japan

**Keywords:** bone tissue engineering, MSCs, osteoblasts, scaffolds

## Abstract

Bone tissue engineering (BTE) is a process of combining live osteoblast progenitors with a biocompatible scaffold to produce a biological substitute that can integrate into host bone tissue and recover its function. Mesenchymal stem cells (MSCs) are the most researched post-natal stem cells because they have self-renewal properties and a multi-differentiation capacity that can give rise to various cell lineages, including osteoblasts. BTE technology utilizes a combination of MSCs and biodegradable scaffold material, which provides a suitable environment for functional bone recovery and has been developed as a therapeutic approach to bone regeneration. Although prior clinical trials of BTE approaches have shown promising results, the regeneration of large bone defects is still an unmet medical need in patients that have suffered a significant loss of bone function. In this present review, we discuss the osteogenic potential of MSCs in bone tissue engineering and propose the use of immature osteoblasts, which can differentiate into osteoblasts upon transplantation, as an alternative cell source for regeneration in large bone defects.

## 1. Introduction

Continuous research is ongoing in bone tissue regeneration technologies related to orthopedics and dentistry. Vast challenges remain, however, in the application of these modalities to reconstituting damaged skeletal structures. Bone grafting has been widely utilized as a regenerative therapy for critical size bone defects (CSDs), and various bone grafting and prosthetic bone materials have been developed in this regard. There is no one standard definition of CSDs. In general, a “critically-sized” defect is regarded as one that would not heal spontaneously within a patient’s lifetime and would require surgical stabilization and further surgical intervention [1,2]. Currently, bone grafting materials are classified as autogenous, allogeneic, or heterogeneous and artificial bone substitutes such as hydroxyapatite (HA), β-TCP (beta-tricalcium phosphate), bioactive glass, and calcium sulfate. Autologous bone has no particular disadvantages other than restrictions on the collection amount and collection site and is recognized as a good prosthetic material with new bone formation capacity. It is thus considered the current gold standard for the regeneration of bone defects but has been most widely used in clinics to treat only small-sized bone defects [3,4,5].

To overcome the limitations of current bone graft therapies, such as autologous bone graft and artificial bone substitutes, many researchers have attempted to develop BTE to regenerate and restore lost bone tissue using MSCs, growth factors, and scaffolds [6,7,8,9,10]. MSCs are referred to as multipotential progenitor cell populations that can differentiate into osteoblast progenitors in vitro under specific conditions, and these cells are most commonly used for bone regeneration [11]. In addition, MSCs are immune tolerant and are used for immunosuppressive therapy via allogenic applications to accelerate bone healing [12]. The use of a scaffold can provide the space needed to deliver and confine MSCs to the bone target site, provide an environment suitable for the migration, proliferation, and differentiation of the stem cells, enable diffusion of nutrients and eventually create early osteoid tissue at the site of the defect which is subsequently mineralized to form new bone. This combination of MSCs and scaffolds has been developed as a BTE therapy. Clinical trials for recovering bone defects have already commenced and reported the accelerated bone healing ability of these approaches. The current bone regenerating ability of the BTE approach is therefore successful but cannot as of yet recover the functional loss caused by large bone defects, such as those resulting from inflammatory diseases.

Osteoblasts are bone-forming cells derived from multipotent mesenchymal stem cells. During skeletal development, multipotent mesenchymal stem cells differentiate into osteoblast progenitor cells and undergo a commitment to form immature osteoblasts that are capable of proliferating before becoming mature osteoblasts. Although mature osteoblasts can synthesize and deposit bone extracellular matrix components, their ability to proliferate is significantly reduced [13]. Thus, recapitulating immature osteoblast differentiation has been suggested as a potential approach to bone regeneration therapy [14]. Previous studies have demonstrated that primary osteoblast cultures from newborns contain large numbers of immature osteoblasts and can expand and heal critical bone defects [15,16]. Currently, osteoblast progenitor cells can be isolated from adult human tissue and are good alternatives to MSCs for bone regeneration. BTE using immature osteoblast and bioscaffolds is, therefore an alternative tissue-engineered construct for recovering large bone defects.

The purpose of this present review is to discuss the importance of MSC-scaffold constructs in BTE, with a particular emphasis on currently available cell sources for clinical translation. In addition, we discuss the bone regeneration potential of transplanted immature osteoblasts as a candidate cell source for BTE.

## 2. MSCs Derived from Embryonic Stem Cells and Induced Pluripotent Stem Cells for Bone Tissue Regeneration

Functional bone tissue engineering generally involves the use of osteoprogenitors derived from MSCs and seeded onto a scaffold to predictably restore the lost architecture and function of bone tissue. MSCs have been isolated from adult tissues such as adipose tissue, bone marrow, and dental tissues, are widely used in regenerative medicine, including BTE, and, thus, have both research and clinical applications. However, MSCs cannot be isolated from patients with systemic disorders such as cardiovascular disease, diabetes, inflammatory bone disease, or advanced aging-related issues. Embryonic stem cell (ESCs)- or Induced pluripotent stem cell (iPSCs)-derived MSCs may be potential cell sources for the clinical trial of BTE [17]. A better understanding of cell fate decisions and differentiation processes during osteoblast development may help to generate functional progenitor cells for tissue restoration. Over the years, technologies involving the osteoblast differentiation of ESCs and iPSCs have been significantly improved, and several studies have demonstrated the successful production of MSCs derived from ESCs/iPSCs for use in BTE therapies [18,19,20] (Figure 1).

### 2.1. ESCs and Bone Regeneration

Human ESCs (hESCs) are pluripotent cells and thus have the potential to form bone-like tissues. However, since ESCs also have tumorigenic potential, they cannot be transplanted directly into bone defect areas and must first be differentiated into either MSCs or osteoblasts for use as a cell source in BTE. Kanke et al. have reported a strategy for mass producing osteoblasts from mouse ESCs using small molecules such as CHIR99021 [CHIR], cyclopamine [Cyc], smoothened agonist [SAG], and a helioxanthin-derivative 4-(4-methoxyphenyl) pyrido [4’,3’:4,5] thieno [2,3-b] pyridine-2-carboxamide [TH] (Figure 1) [18]. Another prior study has established a stepwise protocol to produce an engineered bone graft construct from human ESC-derived mesenchymal stem cell progenitors. This graft material could form mature bone-like tissue upon implantation in immunodeficient mice. These and other previous reports have suggested that an engineered bone construct made using hESCs is a potential graft material for BTE [19]. A recent study by Deng et al. demonstrated that growth differentiation factor 6 (GDF 6) could specifically direct the differentiation of human ESCs into MSCs [21]. In addition, mouse ESCs have previously been successfully seeded onto a ceramic scaffold under a chondrogenic medium to produce a cartilage complex termed tissue-engineered cartilage, which could then form bone-like tissue when implanted subcutaneously into the back of immunodeficient mice and critical cranial defects in the rat [22].

### 2.2. Induced Pluripotent Stem Cells and Bone Regeneration

iPSCs have attracted considerable attention over the past decade and are now considered to be new candidate stem cells for bone regeneration therapy. Various studies have now reported that human iPSCs have similar properties to human ESCs in relation to their morphology, gene expression, differentiation potential, and pluripotency [23]. As explained earlier, osteoblasts can also be obtained from iPSCs in a similar manner to ESCs-derived osteoblasts using small molecules (CHIR, SAG, TH; Figure 1) [18]. In a previous study, researchers performed directed differentiation of mouse iPSC cells to mesenchymal cells and, subsequently, the osteoblast lineage in vitro. These osteoblasts can be seeded onto gelatin scaffolds where they display an osteoblast phenotype, proliferation, and osteogenic matrix production both in vitro and in vivo [24]. Okawa et al. have successfully created iPSCs from mouse gingival fibroblasts and prepared spheroid-like structures that express osteogenic genes and contain an inner unstructured cell mass region surrounded by osteoblast progenitor-like cells, which can induce extensive ectopic bone-like tissue when transplanted subcutaneously into SCID mice [25]. Another recent study has also demonstrated that the bone regenerative ability of calcium phosphate granules (CPG) combined with human iPSCs-derived MSCs (iMSCs) and bone marrow concentrate (BMC) is comparable to that of an autologous bone graft from the iliac crest in a minipig model of surgically-induced critical bone defects in the proximal tibia. These study results demonstrated a significantly higher amount of new bone formation by iMSCs+CPG than from CPG alone, both by histological and computed tomography volumetric analysis. The bone healing ability in the minipigs was found to be comparable between autologous bone grafting and BMC+CPG. Considering the advantages of iMSCs over MSCs and autografts, these cells represent a valuable therapeutic approach to developing new BTE applications [26]. The evidence to date thus suggests that BTE using iPSCs or ESCs represents a promising therapeutic approach to bone regeneration.

Although both ESCs and iPSCs-derived MSCs demonstrate promising results in various preclinical bone defect models, there are still considerable limitations to using these cells in clinical applications, primarily due to patient safety concerns from possible teratoma formation [27,28].

## 3. MSCs for Bone Regeneration

To develop MSCs that have clinical utility for BTE, a standard protocol for the characterization, osteoblast differentiation, and transplantation of these cells in combination with a biodegradable scaffold is required. Various types of MSCs are currently available with osteoblastic lineage differentiation potential; however, their origin and development are not clearly understood. There have been few reports on MSCs being successfully derived from neural crest cells during the development of vertebrates, which is seen as transient embryonic tissue [29]. Most studies on MSCs to date have reported their derivation from perivascular cells, the pericytes. These cells reside in specific niches, which are commonly found in bone marrow, adipose tissue, and various fetal and other adult tissues [30,31]. This has been the primary cell source of the MSCs used in BTE to date.

### 3.1. Characterization of MSCs

Surface markers are currently being used to identify MSCs for quality control assurance prior to cell preparation, based on ‘good manufacturing practice,’ which is required for investor-mediated clinical developments. Hence, the characterization of MSCs based on surface marker analysis is an essential criterion for the clinical application of BTE methodologies. According to the International Society of Cell Therapy (ISCT) criteria, MSCs express a cluster of differentiation (CD) surface markers such as CD90, CD105, and CD73, but do not express CD11b, CD14, CD19, CD34, CD45, or human leukocyte antigen (HLA)-DR [32,33,34]. However, this set of cell surface markers is not always useable for the identification of MSCs. MSCs isolated from different tissues have different surface markers because they are influenced by varying factors. More specific markers are being identified and validated for some particular sources of MSCs. One prior study reported that MSCs expressing CD146 have a higher bone forming capacity and also a homing ability in relation to defective bone sites [35]. Other studies have revealed that CD49d is detectable in adipose-derived MSCs but not in BMMSCs [36]. In another previous study, the authors suggested that CD271-positive MSCs have the highest osteogenic potential with higher induction of osteogenic genes such as DLX5, RUNX2, and BGLAP [37]. It is highly desirable to identify and isolate MSCs with high osteogenic potential, as this will lead to higher efficacy of BTE therapeutics. However, there are no standard techniques or unique properties for detecting a greater osteogenic capacity in these cells, making this a continuing challenge for developing their potential clinical application. Identifying a prospective surface marker that can distinguish the most potent MSCs for use in BTE therapeutics remains a crucial task in terms of establishing the safety and quality control of these cells in clinical translation.

A number of processes play important roles in physiological bone healing, including the immediate responses to a bone defect caused by a fracture or the removal of a necrotic bone tissue/bone tumor, homing, and the recruitment of MSCs. However, critically sized bone defects (CSDs) will not heal on their own without surgical intervention or the use of regenerative therapeutics. Numerous in vitro and in vivo studies have shown to date that transplanted MSCs can have multiple paracrine effects that will enhance bone regeneration through the secretion of trophic factors, immunomodulatory properties of these cells, the recruitment of additional endogenous MSCs, or via the trans-differentiation of these cells into osteoblasts, as shown in Figure 2A [17,38]. These abovementioned effects are the critical properties of MSCs that highlight their essential role in stimulating bone regeneration. A large number of prior articles have extensively investigated and reported on the mechanism of bone regeneration by MSCs and can be referred to for an understanding of the molecular aspects of this process [39,40,41].

### 3.2. Clinical Translation of MSC-Based Bone Regeneration

Over the past decade, a greater understanding of the capabilities of MSCs in BTE has emerged, with numerous preclinical and clinical studies now underway. These trials have revealed the importance of using MSCs in combination with various kinds of scaffolds for the treatment of bone defects and have addressed the future potential to translate this technology to the clinic. To identify suitable cell scaffold constructs for bone regeneration, it is important to review the studies conducted in large animal models and the published results of human clinical trials. The choice of an appropriate animal model that can best approximate human physiology and pathophysiology is critical for the future clinical translation of MSCs to BTE interventions.

#### 3.2.1. BTE Scaffolds

The basic concept behind a scaffold is to mimic the structure and function of the extracellular matrix (ECM) in tissues. The ECM provides both structural and mechanical stability and regulates some of the core cellular functions [42,43,44]. The basic role of scaffolds in BTE is to mimic the ECM of the native bone tissue and provide a functional three-dimensional space for the adhesion, migration, proliferation, and differentiation of osteoblast progenitors in which bone growth can occur [38,45,46,47]. An ideal scaffold for BTE should substitute for both the structure and function of the ECM and thus be capable of regenerating the lost bone tissue when seeded in conjunction with osteoblast progenitors. BTE innovations have led to the development of new biomaterials that resemble the 3D bone structure, in terms of mechanical properties as well as osteoconductive, osteoinductive, and osteogenic features [48,49]. Traditional bone repair approaches mainly focus on the use of bone grafts from autologous, allogeneic, and xenogeneic sources; however, complications such as donor-site morbidity and host immune rejection limit the application of these tissues [50]. The promise of BTE has principally involved overcoming these problems. The aims of BTE are to regenerate and restore the function of lost bone tissue using combinations of osteoblast progenitors and synthetic biomaterial scaffolds. Over the past decade, the use of synthetic biomaterials to enhance bone regeneration has significantly developed because of their capacity to mimic the natural environment of the extracellular matrix. The synthetic scaffold biomaterials predominantly used in BTE include calcium phosphate ceramics, biodegradable polymers, and composites, and the combination of ceramics and polymer scaffolds aims to utilize the properties of both materials [47,50,51].

Various ceramic-based scaffolds have been extensively used in BTE applications to regenerate lost bone and restore function. The most commonly studied of these involve bioactive ceramics such as HA, β-TCP, and biphasic calcium phosphate (a mixture of HA and β-TCP) [51]. Hydroxyapatite (HA) is known for its bioactivity, biocompatibility, nontoxicity, and osteoconductivity. However, although highly biocompatible, HA has unfavorable mechanical properties, as it is brittle and unable to withstand significant compressive loads. The effect of this brittleness is particularly pronounced when using porous ceramic materials. In general, the mechanical strength of porous ceramics decreases drastically with increasing porosity. This is a substantial constraint in the fabrication of porous scaffolds using HA ceramics. More importantly, HA lacks osteoinductivity and true bone regeneration capability [13]. For example, the new bone generated by HA seeded with MSCs has been reported to take the form of a porous HA network that cannot sustain the mechanical load for remodeling [50]. β-TCP bioceramics are quite distinct from other calcium phosphate ceramics such as HA for hard tissue regeneration due to their composition, biocompatibility, degradation, and new bone tissue formation ability [50]. Although the composition of HA is similar to native bone, β-TCP rapidly resorbs compared to HA and becomes replaced with new bone tissue, making it more beneficial than HA as a scaffolding material. However, β-TCP also has unfavorable mechanical properties due to its poor fatigue resistance and brittleness, and these characteristics limit its application as a loadbearing biomaterial [52]. Among the calcium phosphate ceramics, biphasic calcium phosphate (BCP), which comprises different concentrations of stable phase HA and the more soluble phase β-TCP, have presented significant advantages. β-TCP dissolves too quickly to leave an adequate surface area for cell proliferation, and the co-addition of HA aims to control this biodegradation and increase the biological stability of the scaffold [52]. BCP has proven biocompatibility, osteoconductivity, safety, and predictability features based on a number of preclinical and clinical model studies. In the field of BTE, these materials show great promise in the generation of scaffolds capable of carrying and modulating the behavior of MSCs [53]. Overall, ceramic scaffolds are more commonly used in BTE applications, owing to their similar chemical structure and composition to natural bone, along with their bioactivity, osteoconductivity, and osteoinductivity properties.

Polymers represent another key material that has been investigated in the fabrication of suitable scaffolds for BTE. Scaffolds made of polymeric biomaterials typically provide good structural support for cell attachment and subsequent tissue development. Biodegradable polymeric materials can be categorized as natural or synthetic. The naturally-derived polymers that are commonly used in BTE applications include collagen and gelatin (derived from collagen), but these materials are limited by their instability, incompatible characteristics, immunogenicity, and poor biodegradability [47]. Synthetic polymers, on the other hand, can exhibit excellent results in BTE due to their thermo-modifiable properties. Synthetic polymer-based scaffolds made of polyglycolic acid, polylactic acid, polycaprolactone, and copolymers are commonly used polyesters in BTE. Importantly, their degradation products in the human body can be removed by natural metabolic pathways [54]. The main advantages of synthetic scaffolds are the ability to custom design them for the defect area, their higher mechanical properties, and the capacity to control the scaffold micro-architectures such as the pore size and its distribution, and to regulate the biodegradability rate [55,56]. Although many factors could be modified to fabricate a bioscaffold appropriate for BTE applications, pore size and interconnectivity of the pores are critical scaffold parameters that greatly influence the characteristics and amount of new bone formation [57]. Scaffold porosity and pores size are vital for the diffusion of nutrients and clearance of wastes, provide adequate mechanical stability to support and transfer mechanical loads, and appropriate material surface chemistry to allow cells to express their normal phenotype for bone regeneration [58,59]. Pore size for the scaffolds has been extensively investigated to identify the optimal range for bone tissue regeneration. Recent work by Lee et al. compared the effects of pore size (250 and 500 μm) of hydroxyapatite collagen-based scaffold (HCCS-PDA) on the regeneration of large bone defects. The results showed a limited amount of new bone formation in the 250 μm pore scaffold, and, in contrast, a more significant amount of new bone was seen in the 500 μm pore scaffold [60]. According to Cheng et al., magnesium scaffolds with two pore sizes of 250 and 400 μm, the larger pore size leads to a more significant amount of new bone by enhancing angiogenesis. This study concluded that larger pore size promoted early vascularization and up-regulated collagen type 1 and osteopontin expression, resulting in greater bone mass and more mature bone formation [61]. Similarly, several other studies have attempted to determine the optimal pore size for bone regeneration and found it to be in the range of 100–700 μm [62,63]. Despite numerous similar studies, no clear consensus has emerged with respect to the optimum pore size for bone regeneration. However, notwithstanding these characteristics of individual synthetic polymers that lead to improved osteoconductivity in BTE applications, current trends involve combining these compounds to produce a scaffold. Recently, composites of bioceramic scaffolds comprising polymers such as poly (lactic-co-glycolic) acid (PLGA), which mimic mineral component and microarchitecture of native bone tissue, were developed to increase mechanical stability and improve tissue interactions. The infiltration of polymers such as PLGA results in significantly enhanced mechanical properties compared to non-infiltrated TCP scaffolds, thereby balancing the issues with brittleness [47]. Hence, microstructured inorganic composites of HA and biodegradable polymers, such as collagen, gelatin, chitosan, and polylactic acid, offer an alternative solution to some of the earlier mentioned drawbacks. In one recent study, a synthetic biodegradable composite scaffold, comprising ceramic tri-calcium phosphate (TCP) infiltrated with polymer poly (D, L-lactide-co-glycolide (PLGA), was seeded with MSCs and found to significantly improve bone regeneration in large mandibular defects [50]. A study by Moncal et al. demonstrated significant healing of rat critical-sized calvarial bone defects following the transplantation of hybrid scaffolds (collagen infilled 3D printed synthetic polymers) combined with transfected rat BMMSCs than the control group [62]. To further demonstrate the potential use of hybrid scaffolds in BTE applications, Dong et al. and their co-workers investigated whether chitosan/PCL scaffolds could improve the proliferation and expedite osteogenesis of BMMSCs compared to individual components of the hybrid scaffold. The results showed that chitosan/PCL hybrid scaffold is favorable for cell survival, even cell distribution, better cell retention, and enhanced osteogenesis compared to chitosan or PCL alone [63].

Although material science technology has resulted in major developments in the field of bone regeneration, no scaffold material has yet been developed that can achieve the complete regeneration of critical sized bone defects (CSDs), and this still remains a major challenge. Human bone is a composite of HA representing the ceramic phase and collagen forming the polymer phase. Hence, the development of hierarchical porous scaffolds to imitate the structure and properties of the natural human bone is a significant issue in tissue engineering. Although bioactive ceramics are regarded as promising biomaterials due to their comparable chemical compositions to human bone, their low strength limits their biomedical application with respect to load-bearing sites. On the other hand, polymeric biomaterials feature some disadvantages such as insufficient mechanical strength, low bioactivity, and lack of cell adhesion binding sites. Hence, composites are of great importance as they combine the excellent ductility of biopolymers and the bioactivity of ceramics. Although numerous types of scaffolds are now commercially available, ceramic scaffolds using HA and calcium phosphate-based materials are the most commonly used for BTE because of their higher osteoconductive properties and biocompatibility.

#### 3.2.2. Preclinical Studies of BTE in a Large Animal Model Using MSC/Scaffold Combinations

To translate the clinical use of MSCs combined with scaffolds for BTE, large animal model systems that closely resemble human physiology are required. A number of preclinical studies conducted using MSCs with varying combinations of biomaterials in critical bone defect models are listed in Table 1.

In addition, the results from the use of MSCs seeded with scaffolds to regenerate large animal bone defects provide highly relevant evidence to assist with future clinical applications in human patients. There have been 10 relevant studies in the literature, 8 in pigs and 2 using monkeys, which evaluated bone tissue regeneration using cell-scaffold constructs (Table 1). Six of these studies used BMMSCs [64,65,66,67,68,69], two studies Adipose derived MSCs (ADSCs) [50,70], one study gingival MSCs [71] and another study periodontal ligament stem cells (PDLSCs) [72] as the cell source. Bioactive ceramic-based scaffolds such as β-TCP [69], calcium phosphate cement [66], HA/TCP (hydroxyapatite/tricalcium phosphate) [68], and bioactive glass [64] have been the most commonly used scaffolds for BMMSCs in BTE. Wang and co-workers demonstrated that BMMSCs seeded with bioactive glass in conjunction with the BMP2 (bone morphogenic protein 2) gene showed faster healing, successfully recruited endogenous MSCs and induced the differentiation of implanted MSCs, and promoted the rapid recovery of critical alveolar bone defects [64]. Qiu et al. demonstrated that the co-delivery of BMMSCs and platelet rich plasma (PRP), seeded with calcium phosphate cement (CPC), significantly increased new bone and blood vessel formation compared to CPC alone in a large femoral condylar defect model using a minipig [66]. The other studies listed in Table 1 demonstrate the effectiveness of BMMSC-ceramic scaffold constructs in bone regeneration using critical size bone defect models compared to scaffolds alone or scaffolds combined with exogenous factors without cells. The results of these studies further highlight the importance of cell-scaffold constructs in bone regeneration. In the majority of the preclinical studies reported to date, the role of additional factors such as osteogenic inducers, transfection with particular genes, or the incorporation of PRP can enhance bone regeneration when using cell-scaffold constructs.

MSCs have an important role in a wide range of therapeutic applications, including bone regeneration of CSDs; however, it is difficult to ascertain the specific role of transplanted cells in the regeneration of bone defects. Implanted cells face many challenges starting from culturing until transplantation, including length and duration of culture conditions, mechanical stress during implantation, reduced oxygen and nutrient supply for their survival, proliferation, and differentiation. On the contrary, there are many studies available that demonstrate the fate of implanted cells and their contribution to the regenerative outcome. A study by Lalande et al. demonstrated the survival of transplanted adipose-derived stem cells labeled with magnetic agents within a three-dimensional porous polysaccharide scaffold by magnetic resonance imaging (MRI) until 28 days after implantation subcutaneously in nude mice [73]. Brennan and co-workers reported cell fate and the biological role of transplanted cells, including cell density within the biomaterial following transplantation into a critical size bone defect and ectopic site. Their study showed that increasing cell density did not significantly yield more bone regeneration, and only 1.5% of transplanted cells remained after five weeks of implantation. The main reason for cell death is the hypoxic environment and reduced glucose for BMMSCs at the implant site. Despite significant cell loss, a higher amount of bone regeneration was observed in the seeded BMMSCs biomaterial group. This effect is mainly due to recruiting host BMMSCs, suggesting that transplanted BMMSCs release paracrine factors that play an essential role in new bone formation [74]. In a more recent study, Hsieh and co-workers compared seeded and host cells’ distribution and proportion by tracking two fluorescent cells in the same scaffold in a transgenic domestic pig critical-sized calvarial defect model. The results from both in vitro and in vivo experiments showed that the seeded cells were present until four weeks. Also, they concluded that seeded cells recruit host cells and contribute to significantly higher bone regeneration than that of the control group (scaffold without cells), indicating that seeded cells play a critical role in the osteogenic differentiation process [65]. Although the bone regeneration by MSCs was initially thought of due to their ability to differentiate into multiple cell lineages once engrafted in the recipient tissue, nowadays, numerous studies have found that the extensive secretion of paracrine factors from MSCs appears to be related to the therapeutic action of MSCs [62,74,75]. This topic remains the subject of considerable research.

#### 3.2.3. Gene Therapy for Bone Regeneration

Gene therapy is another promising approach for enhancing bone regeneration. Today, the advancement in life-sciences technology allows gene transfer technology to fabricate a tissue-engineered scaffold to accommodate the growth of genetically modified cells and the endogenous synthesis of desired gene products in a controlled manner. Gene therapy allows for the transfer of genetic material in the precise anatomic location of target cells, allowing the transgene expression from the cells with the currently available techniques [76]. Gene transfer can be performed by several ex vivo and in vivo delivery techniques and by either using viral (transduction) or non-viral (transfection) vectors [76,77]. Since this review is focused on the use of combined cell scaffold constructs for bone regeneration, we mainly discuss the ex vivo delivery method, which requires isolation of target cells and transfer of the desired gene to express the respective protein in vitro and then seeded onto the biocompatible carrier material to obtain cell-scaffold construct for bone tissue engineering applications. The two standard methods of ex vivo delivery include viral and non-viral, it being said that each type has its advantages and disadvantages. Viral vectors demonstrate high transfection efficiency with immunogenicity and toxicity, raising an issue of safety. In contrast, non-viral vectors usually consist of plasmid or related DNA, which are non-immunogenic and high safety but with low transfection efficiency [77,78]. Another promising approach is the sequential delivery of exogenous genes to promote the osteogenesis of stem cells. For example, genes that are expressed early and in the final stages of osteogenesis are different. Hence, delivering required osteogenic genes at specific time intervals into target cells induces efficient osteogenic differentiation. A recent study by Kim et al. demonstrated an effective sequential delivery of runt-related transcription factor 2 (RUNX2) and osterix genes induced conversion of human MSCs into pre-osteoblasts and subsequent delivery of activating transcription factor 4 (ATF4) gene triggered further osteogenesis. Differentiation of MSCs into desired mature cells can be regulated by the delivery time of specific osteogenic genes mimicking the natural process of bone remodeling [78,79].

Proof of principle has been established using small animal models such as mice, rats, or rabbits using a variety of different transgenes, including those encoding morphogens (BMPs, hedgehog proteins), growth factors (PDGF, FGFs, IGFs), angiogenic factors (VEGF isoforms and FGFs), and transcription factors (Runx2, Osterix, Sox9), targeting gene therapy-based bone regeneration [78,80,81,82]. In addition, due to increased understanding of the molecular basis of bone remodeling and gene therapy, RNA (including messenger RNA (mRNA), microRNA (miRNA), and short interfering RNA (siRNA)) based therapeutic approaches have recently gained significant attention for bone tissue engineering [83]. A recent study by Moncal et al. demonstrated the effective repair of critical-sized calvarial bone defects utilizing miRNA-based therapy. A small number of studies demonstrate the efficacy of gene therapy for bone regeneration in large animal models. For example, in one study, BMMSCs were engineered with the adenovirus expressing BMP7 (AdBMP7), seeded into coral scaffolds, and implanted into the critical-size femoral defect in the goat model. The study results revealed that BMP7 gene-modified BMMSCs promote greater healing than the non-transduced group [84]. Another study by Lin and co-workers investigated genetically engineered adipose-derived stem cells (ASCs) using baculoviruses to express BMP2/VEGF on massive bone healing in minipigs. In this study, transduced ASCs combined with apatite-coated PLGA scaffolds promoted remarkable complete healing of the bone defect compared to a mock traduced group, indicating the potential of gene therapy-based bone tissue engineering for future translational research [70]. Although the ex vivo delivery method is safer and allows for the identification of any abnormalities before implantation and checking expression levels of the desired genes, it is technically more demanding [76]. Despite the promising results from preclinical studies, especially BMPs, using gene therapy for bone tissue engineering, efficacy, and biological safety need to be thoroughly investigated in large animal models such as pig, sheep, and goats before being implemented in the clinical trials.

Overall, the positive results of the aforementioned preclinical studies using large animal models can be attributed to the combined effect of BMMSCs and ceramic scaffolds, which possess structural similarities to the mineral phase of bone and also have osteoconductive properties. Thus far, several large experimental animal models have revealed the regeneration potential of MSCs in conjunction with various scaffolds. The majority of these prior animal studies have indicated that the combination of BMMSCs with calcium phosphate ceramic scaffold material has a significantly beneficial impact on bone regeneration and function.

#### 3.2.4. Clinical Trials of MSCs for BTE

Over the past decade, a greater understanding has emerged with regard to the capabilities of MSCs to promote bone tissue regeneration, with numerous preclinical and clinical studies now underway. To identify the current potential combination of cell-scaffold constructs or tissue-engineered substitutes for bone tissue regeneration, we found twenty clinical trials. Nine are published (Table 2), and others are listed in the ClinicalTrails.gov database (Table 3). These trials have highlighted the importance of using cell-based therapy with various scaffolds to treat bone tissue regeneration in a real clinical setting. From the twenty identified clinical studies listed in Table 2 and Table 3, the majority report the use of BMMSCs, reflecting the fact that they are the most accepted cell source and the current gold standard in most clinical trials for treating bone disease, including nonunion fractures of long bones and craniofacial bone defects. However, in a few clinical trials, researchers have used umbilical cord (UC)- MSCs [85], BMMSCs [86], and adipose-derived MSCs as allogeneic cell sources to prepare the tissue-engineered constructs for regeneration of critical bone defects (NCT02307). Ceramic-based scaffolds are the primary choice in the majority of clinical trials, indicating their high clinical relevance. From the clinical trials listed in Table 2 and Table 3, most studies used a combination of BMMSCs with calcium-phosphate ceramics such as hydroxyapatite [85,87,88], β-TCP [86,89] (NCT02803177, NCT02153372), biphasic calcium phosphate, a combination of hydroxyapatite and β-TCP [90] (NCT04297813, NCT03325504, NCT01842477). Although most of these clinical trials used a simple combination of calcium-phosphate ceramics with BMMSCs, in a few studies, however, additional factors were included to facilitate enhanced bone regeneration. For example, Dilogo et al. added growth factor BMP2 along with cell scaffold constructs to enhance bone regeneration [85,88]. Similarly, researchers used BMMSCs mixed with BMP2 and loaded them on to 3-dimensional tissue-engineered collagen scaffold (NCT01958502) in another clinical trial. However, a clinical study by Baba et al. used polylactic scaffold seeded BMMSCs mixed with platelet-rich plasma solution and an additional 5000 units of human thrombin dissolved in 10% calcium chloride [91].

The approaches of the trials described to date in the literature to increase bone regeneration at defect sites using BTE have been: (1) MSCs combined with ceramic based scaffold material and (2) MSCs combined with ceramic based scaffold material with the inclusion of additional factors such as platelet rich plasma or growth factors such as BMP2. However, although the above two approaches for BTE have been most widely utilized, no standardized procedures have yet been established for preparing tissue engineered products for transplantation into critical bone defects. This is because several factors have differed between previous studies of BTE methods such as the MSC source and isolation technique, choice of scaffold (with or without growth factors), and various steps used in the preparation of the tissue engineered construct.

However, there is no definite set of standard rules for the preparation of clinical-grade cell-scaffold constructs with a preserved capacity to regenerate new bone for the treatment of various CSDs. Here, we describe the standard method for preparing biological cell-scaffold constructs based on the clinical trials listed in Table 2 and Table 3. For autologous MSCs transplantation, bone marrow aspirates were harvested from posterior iliac crest [88,90,91,92] and, in cases of allogeneic transplantation, adipose or umbilical cord tissue were collected [85] (NCT02307). Harvested tissues were screened for contamination and placed in a sterile container kit, then shipped to certified GMP grade cell factory for further tissue processing. On arrival, cell count and viability were tested for bone marrow aspirates before any manipulation and they considered inadequate if the white blood cell count was not in the normal range [90]. Harvested tissues are processed for cell isolation, culturing, and expansion using clinical-grade reagents under laminar hood flow in grade A clean room conditions [90]. All the isolated cells were tested for appropriate quality control sterility checkpoints free from contamination, bacterial endotoxins, and mycoplasma aerobic/anaerobic pathogens at the beginning, middle, and end of the cell culture [90]. In the case of allogeneic transplantation, culture expanded cells are aliquoted, cryopreserved, and stored in a cell bank under liquid nitrogen until their use [86]. The expanded MSCs (for autologous use) or cryopreserved MSCs (for allogenic use) are subsequently characterized by demonstrating MSCs phenotype [86], typical MSCs surface markers by flow cytometry [85,88], osteogenic differentiation ability [86,91], cell number, cell viability by trypan blue [88,90], and cytogenetic abnormalities [86] prior to release for clinical use. The desired number of cells are either combined with appropriate biomaterial within the GMP facility or only cell suspension diluted with saline and dispatched to the clinical trial unit wherein cells are mixed with appropriate scaffold and inserted into the implant site [85,86,90]. The schematic of standard procedure in accordance with good manufacturing practice for preparation of cell-scaffold construct for BTE applications is shown in Figure 3.

**Table 2 cells-10-02687-t002:** Completed and published clinical studies using MSCs combined with biomaterials for bone tissue regeneration.

Author	Type and Size of Defect	Transplant Groups	Origin of Cell Source	Pre-Transplant Incubation	Outcome
Dilogo et al., 2020 [85]	Nonunion fractures of Humerus/tibia with critical size bone defects	Combination of HA Bongros^®^-HA, Daewoong), BMP2, UC-MSCs with demineralized bone matrix	Allogeneic Umbilical Cord MSCs (UC-MSCs)	None	Allogeneic UC-MSCs can be used safely to treat the critical sized bone defects of long bones.
Dilogo et al., 2019 [88]	Humerus, Tibia and Femur Critical sized defects	Combination of HA granules (Bongros^®^-HA, Bioalpha, Seungnam, Korea), BMP2 and BMMSCs mixed with Plasma solution.	Autologous Bone marrow harvested from posterior Iliac crestal bone	None	Dramatic improvement of bone regeneration compared to preoperative radiographs.
Gjerde et al., 2018 [90]	Severe mandibular ridge resorption.	Expanded, autologous MSCs with biphasic calcium phosphate (MBCP^+^TM; Biomatlante, France)	Bone marrow cells from the posterior iliac crest	None	MSCs successfully induce significant new bone formation
Baba et al., 2016 [91]	Intrabony Periodontal defect. Probing depth >4 mm	The mixture of BMMSCs and PRP, combined with human thrombin dissolved in 10% calcium chloride perfused in a 3D woven-fabric composed of poly-L-lactic acid resin fibers (MSCs/PRP-3D woven Fabric)	Autologous Bone marrow harvested from posterior Iliac crestal bone	Induced under Osteogenic Medium	BMMSCs/PRP-3D woven Fabric constructs showed efficient regeneration of the periodontal tissue including alveolar bone.
Morrison et al., 2018 [86]	Cranial defects with less than 80 mm diameter	Allogeneic mesenchymal stromal cells (MSCs) on a ceramic carrier (ChronOS granules, synthes, and polymer scaffold,	Allogenic BMMSCs from 18–25 years aged donors	None	Allogeneic MSCs can be safely used for bone regeneration.
Kaigler et al., 2015 [89]	Severe Bone Atrophy of upper Jaw	Combination of BMMSCs and β-TCP (Cerasorb, Curasan AG, Germany)	Autologous Bone marrow harvested from posterior Iliac crestal bone	None	Higher density of regenerated bone with MSCs+ β-TCP group was observed than control group.
Marcacci et al., 2007 [87]	Humerus, Tibia and ulnar Critical sized defects	Combination of invitro expanded BMMSCs seeded with porous hydroxy apatite scaffolds (Finblock, FinCeramica Srl, Faenza, Italy)	Autologous Bone marrow harvested from posterior Iliac crestal bone	None	Significant healing of the CSDs. Attained long term durability of bone regeneration.
Bajada et al., 2007 [93]	Tibial non-union	Combination of invitro expanded BMMSCs seeded with calcium sulphate pellets (Stimulan, Biocomposites Ltd., Keele, United Kingdom)	Autologous Bone marrow harvested from posterior Iliac crestal bone	None	Clinical and radiological healing of nonunion was observed
Morishita et al., 2006 [94]	Tibial/femur massive defects	Attachment of invitro expanded BMMSCs-HA granules	Autologous Bone marrow harvested from posterior Iliac crestal bone	Induced under Osteogenic Medium	Good integration of BMMSCs-HA constructs to the host bone and increased radiographic density of the defect area.

**Table 3 cells-10-02687-t003:** List of clinical studies listed in clinicaltraisl.gov using MSCs combined with biomaterials for bone tissue regeneration.

NCT Number	Brief Title	Phase	Conditions	Interventions
NCT04297813	Efficacy in Alveolar Bone Regeneration With Autologous MSCs and Biomaterial in Comparison to Autologous Bone Grafting	Phase I	• Alveolar Bone Atrophy	Autologous MSCs and a biomaterial, biphasic Calcium Phosphate (BCP).
NCT03325504	A Comparative Study of 2 Doses of BM Autologous H- MSC+Biomaterial vs. Iliac Crest AutoGraft for Bone Healing in Non-Union	Phase III	• Non Union Fracture	Culture-expanded autologous BMMSC combined with biphasic calcium phosphate (BCP) biomaterial granules
NCT02803177	Cell Therapy by Autologous BMC for Large Bone Defect Repair	Phase II	• Humerus Fracture Displaced Proximal	Autologous Bone Marrow-derived Mononuclear Cells (BMC) seeded onto ß-TCP
NCT02307435	Allogenic Mesenchymal Stem Cell for Bone Defect or Non Union Fracture	Early Phase I	• Non Union Fracture, Metaphyseal Fibrous Defect	Allogeneic MSCs from umbilical cord/bone marrow/adipose combined and HA-CaSo4
NCT02153372	Cell Therapy by Bone Marrow- derived Mononuclear Cells (BMC) for Large Bone Defect Repair: Phase-I Clinical Trial	Phase I	• Humerus Fracture Displaced Proximal	Autologous Bone Marrow-derived Mononuclear Cells (BMC) seeded onto ß-TCP
NCT01958502	Evaluation the Treatment of Nonunion of Long Bone Fracture of Lower Extremities (Femur and Tibia) Using Mononuclear Stem Cells from the Iliac Wing Within a 3-D Tissue Engineered Scaffold	Phase II	• Nonunion of Fracture	BMMSCs with BMP2 within a 3-D tissue engineered collagen scaffold
NCT01842477	Evaluation of Efficacy and Safety of Autologous MSCs Combined to Biomaterials to Enhance Bone Healing	Phase I/II	• Delayed Union After Fracture of Humerus, Tibial or Femur	BMMScs mixed with biphasic calciulm granules
NCT00250302	Autologous Implantation of Mesenchymal Stem Cells for the Treatment of Distal Tibial Fractures	Phase I/II	• Tibial Fracture	BMMSCs loaded onto a carrier and implanted locally at the defect site
NCT00557635	Osseous Setting Improvement With Co-implantation of Osseous Matrix and Mesenchymal Progenitors Cells From Autologous Bone Marrow	Phase II	• Tibia or Femur Pseudo-arthrosis	Injection of an osseous matrix (osteopure) combined with MSC progenitors from autologous bone marrow.
NCT02177565	Autologous Stem Cell Therapy for Fracture Non-union Healing	Not available	• Non-union of Fractures	Autologous BMSCs combined with carrier material
NCT01435434	Mononucleotide Autologous Stem Cells and Demineralized Bone Matrix in the Treatment of Non Union/Delayed Fractures	Not available	• Non Union/Delayed Fractures	Injection of Autologous Stem Cells and Demineralized Bone Matrix

Culture-expanded MSCs combined with ceramic-based scaffolds have been reported in several clinical trials to have no long-term follow-up complications. In one recent trial, autologous culture-expanded BMMSCs were seeded onto hydroxyapatite granules mixed with BMP2 to treat CSDs in long bones. In that study, the selected CSDs patients had previous surgical failures to fix the defect. Local implantations at the defect site of 5 × 10^7^ BMMSCs resulted in a dramatic improvement in bone graft incorporation over a 12 month follow-up period, which was a promising result in terms of fracture healing and scaffold integration in the host [88]. In another relevant study, Marcacci et al. utilized in vitro expanded MSCs to treat 4–7 cm bone defects in six patients. The porous hydroxyapatite-tri-calcium-phosphate (HA-TCP) scaffolds used in these cases were designed to match the size and shape of the defect. Complete fusion and integration of the scaffold and host bone were achieved after 5–7 months, which were further promising results for the repair of CSDs using BTE. The HA-TCP constructs used demonstrated superior effects with regard to cell proliferation, calcium deposition, and collagen bundle formation [87]. Bajada et al. successfully treated a nine-year tibial nonunion that had been resistant to six previous surgical procedures using autologous bone marrow stromal cells that were expanded to 5 × 10^6^ cells combined with calcium sulfate (CaSO4) pellets after three weeks of tissue culture [93]. Morishita et al. treated massive defects at the distal tibia following tumor resection by transplanting autologous BMMSCs and HA granules. In that study, BMMSCs-HA constructs were cultured for two weeks each under basal and osteogenic medium before use at the defect area. Weight-bearing became possible two weeks after transplantation in two patients and after three weeks in one further patient. The CT images revealed that the cultured tissue engineered constructs showed good integration with the host bone after three months and that the radiographic density increased eventually without any adverse effects after long follow periods of two to four years [94].

Several other studies have also investigated the use of in vitro expanded MSCs to regenerate bone [86,89,90,91]. Based on limited and heterogeneous evidence from clinical studies, BMMSCs in combination with a ceramic-based scaffold appear to result in an efficacious tissue engineered cell-scaffold construct for bone regeneration by recapitulating the in vivo bone microenvironment. However, further studies are still required to build on the current preclinical and clinical evidence for BTE to address limitations in facilitating tissue and site-specific osseous repair. Nevertheless, a significant number of clinical trials are currently being conducted with a promising degree of success, further supporting the potential to combine MSCs and scaffolds for successful BTE applications (Table 2).

## 4. Osteoblast-Based Bone Tissue Regeneration

In addition to the efforts to increase the bone-forming ability of MSCs as a cell source for bone tissue engineering, the use of osteoblasts that are capable of proliferating before maturing, and that can synthesize and deposit bone extracellular matrix components such as osteocalcin (OCN) and bone sialoprotein (BSP), provides a potential alternative BTE cell source for the treatment of large bone defects. However, since BTE is generally approached using a combination of osteoblasts induced from MSCs on biodegradable scaffolds, the resulting bone forming efficacy will be dependent on the differentiation potential of MSCs into osteoblasts. This could hamper the progress of BTE for treating large bone defects. There are two major mechanisms underlying skeletal development, intramembranous and endochondral ossification. In intramembranous ossification, osteoblast lineage cells, i.e., immature osteoblasts, are formed directly from condensed mesenchymal tissue. Endochondral ossification, by contrast, involves the production of osteochondral progenitors from MSCs that give rise to hyperchondrocytes which activate perichondrial cells to differentiate into immature osteoblasts. From the perspective of BTE, the formation of immature osteoblasts is the convergence point for both types of ossification.

### 4.1. Development of Immature Osteoblast-Based BTE

BTE using immature osteoblasts derived from the human maxilla was conducted previously in nude rats using two different biomaterials, polyhydroxybutyrate embroidery and hydroxyapatite collagen tape. The results of that study revealed the induction of ectopic bone formation using either of these biomaterials [95]. Ortiz et al. evaluated the proliferation and calcium phosphate deposition ability of primary human osteoblasts seeded onto a 3D polyglycolic acid scaffold functionalized with the RGD (R: arginine; G: glycine; D: aspartic acid) peptide (PGA-RGD). The results of that investigation revealed that 92–98% of the seeded cells survived with significantly higher proliferation and mineralization levels on PGA-RGD compared with the control group (PGA) [96]. These data indicated that osteoblasts grown on 3D polymeric scaffolds can be used for BTE. In another recent study, the adhesion and viability of immature human osteoblasts were investigated on different tridimensional structures fabricated from hydroxyapatite, collagen, porous silica, and bovine bone. All of these materials provided a compatible surface for cell adhesion and viability. However, better adhesion was observed with bovine bone and a higher viability was evident when using a collagen scaffold. The results of that study thus suggested that all of these materials can be used with osteoblasts as a scaffold material for bone regeneration in both the medical and dental field [97].

The isolation of human immature primary alveolar osteoblasts (HAOBs) from young and middle-aged donors using a defined culture medium by collagenase enzymatic digestion was established previously as a standard protocol. These cells have also shown a comparable proliferative capacity, whether derived from young or middle-aged donors. Moreover, HAOBs obtained via this methodology exhibited significantly higher osteogenic ability than MSCs, either in in vitro or in vivo [14]. More importantly, HAOBs have demonstrated bone-forming ability upon transplantation into immunodeficient mice, suggesting that they are suitable for bone regeneration therapy by autologous transplantation. HAOBs have also shown high nebulette (NEBL) expression, an actin-binding protein, during their ex vivo expansion and have high osteogenic potential. The gene knockdown of *NEBL* inhibits mineralized nodule formation, alkaline phosphatase activity, and the expression of bone marker genes, indicating that it can be used as a functional qualitative HAOB marker in the development of bone regenerative therapeutics with these cells. However, further studies are needed on an appropriate biodegradable scaffold with HAOBs for bone regeneration using large animal models such as pigs or non-human primates for future translational studies. This will facilitate to develop clinical protocols for regeneration therapies of large bone defects using these cells.

Based on the current evidence in the literature, bone regeneration in larger defects now appears to be feasible via the transplantation of an immature osteoblast-seeded bioscaffold (Figure 4A). The immature osteoblasts seeded onto this biocompatible scaffold will proliferate and differentiate into mature osteoblasts producing bone matrix components and angiogenesis factors [98]. Most osteoblasts become embedded inside the bone matrix to become osteocytes; however, some others remain as bone lining cells on the outer surface. Simultaneously, when osteoblasts lay down a new matrix, osteoclasts will differentiate from circulating monocytes/macrophages. During these processes, the differentiation of immature osteoblasts is regulated by several cytokines, including bone morphogenetic proteins (BMPs), which can strongly promote osteoblast differentiation [75]. BMPs belong to the transforming growth factor family, which can transduce signaling activity through specific type I and type II transmembrane kinase receptors. After BMPs bind to these receptors, the type II receptor binds to the type I receptor, then activates the type I receptor to phosphorylate Smad 1/5/8. This phosphorylation of SMAD1/5/8 causes it to form a complex with SMAD4 in the cytoplasm that then translocate to the nucleus to regulate the expression of osteoblast marker genes such as ALPase, OCN, and BSP. BMP2 also controls the expression of RUNX2 and OSX, which are essential transcription factors for osteoblast differentiation. Several groups have reported that VEGF is abundantly expressed by osteoblastic cells of mouse, rat, and human origin and is regulated by the hypoxia inducing factor (HIF) signaling pathway. In this regard, preclinical studies have shown that increased HIF activity in osteoblasts or endothelial cells promotes angiogenesis and bone formation. Interestingly, VEGF derived from osteoblasts or released from the resorbed matrix can also stimulate osteoclast formation [99,100]. Osteoblasts also express RANKL, a member of the tumor necrosis factor (TNF) family. RANKL is, in fact, a TNF superfamily member which can play significant roles in the regulation of osteoclast differentiation via cotreatments with bone resorption-stimulating factors such as 1α,25- dihydroxy vitamin D3 [1α,25(OH)2D3], parathyroid hormone, and prostaglandin E2 (Figure 4B) [100].

Although the aforementioned studies indicate that immature osteoblasts promote bone regeneration through their differentiation and subsequent promotion of angiogenesis and osteoclastogenesis, the effective induction of functional osteoblasts on biodegradable scaffolds upon implantation will require further investigations to develop BTE approaches for large bone defects.

### 4.2. Clinical Trial of Osteoblasts for BTE

Immature osteoblasts can be isolated from adult skeletal tissue, including the maxilla and mandible during periodontal surgery [101], from the hip bone during a hip arthroplasty [102,103], and also from defects at sites such as the iliac crest or femoral head during surgical reconstruction [104]. These bone tissue samples are dissected into small pieces and are either kept in dishes for explant cultures or digested enzymatically for ex vivo expansion. To date, few clinical studies have described the use of primary osteoblasts as a cell source for BTE treatments of critically sized bone defects. The regenerative outcome obtained from these studies needs to be carefully determined due to insufficient characterization of immature osteoblasts prior to the transplantation [105,106,107].

## 5. Conclusions

Critical bone defects that cannot self-heal without a surgical intervention pose a significant challenge in the field of BTE. Compared to the traditional gold standard approach of using autogenous bone, regenerative methods will typically use either exogenous MSCs or immature osteoblasts seeded onto a bioactive scaffold placed at the defect area to regenerate functional bone. Adult MSCs from bone marrow and adipose tissue have often been used in various clinical studies for bone regeneration. The available data from both preclinical studies and clinical trials have shown promising results when BMMSCs are used as a cell source for bone tissue regeneration. Many clinical studies have also shown the beneficial effects of MSCs and scaffold combinations in bone healing. Although encouraging clinical results have been obtained by transplanting MSCs-scaffold constructs, the exact dosage and route of application remains to be optimized, and the fate of transplanted cells and their mechanisms of action need to be better monitored in more extensive future clinical trials. The development of an alternative immature osteoblast source combined with more effective scaffolds is also anticipated in the future. Immature osteoblasts have the ability to become a potential alternative cell source to adult MSCs, as they are osteogenic lineage-committed cells that enhance the efficacy of bone regeneration.

Immature osteoblasts can be obtained from bone tissue samples during routine oral surgical procedures from mandible/maxillary alveolar bone or surgeries involving long bones from the femur or tibia. The advantage of immature osteoblasts over MSCs is their spontaneous matrix formation upon transplantation without promoting osteogenic differentiation. Immature osteoblasts can directly secrete bone collagen matrix and release various factors such as M-CSF, RANKL, and VEGF, enhancing bone remodeling and bone regeneration in various BTE applications. The combination of an immature osteoblast culture system which possesses robust osteogenic activity and an appropriate biodegradable scaffold is an expected future BTE therapeutic option. This approach will facilitate the establishment of better clinical protocols for regeneration therapy in cases of large bone defects, as a treatment for orthopedic conditions such as back pain resulting from stenosis and lumbar spondylolisthesis osteosarcoma, and for horizontal alveolar bone defects in the dental field.

## Figures and Tables

**Figure 1 cells-10-02687-f001:**
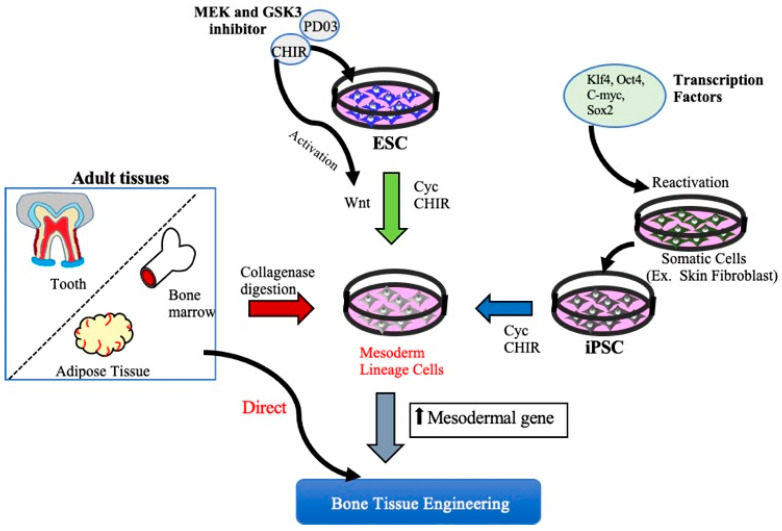
Schematic diagram of different approaches to obtain Mesenchymal stem cells. MSCs can be derived from either iPSCs, ESCs, or adult mesenchymal tissue. MSCs can be obtained by ESCs and iPSCs using small molecules such as mitogen-activated protein kinase (MEK) inhibitor, (MEK) inhibitor, PD0325901, glycogen synthase kinase 3 (GSK3) inhibitor, and CHIR99021 (CHIR). MSCs are also be derived from various connective tissues such as bone marrow, adipose tissue, and dental tissues by collagenase digestion or aspirates from bone marrow and adipose tissue directly used for BTE therapeutics. KLf4: Kruppel Like Factor 4, Oct4: Octamer-binding transcription factor 4, C-myc: Cellular-Myelocytomatosis, Sox-2: sex-determining region Y-box 2.

**Figure 2 cells-10-02687-f002:**
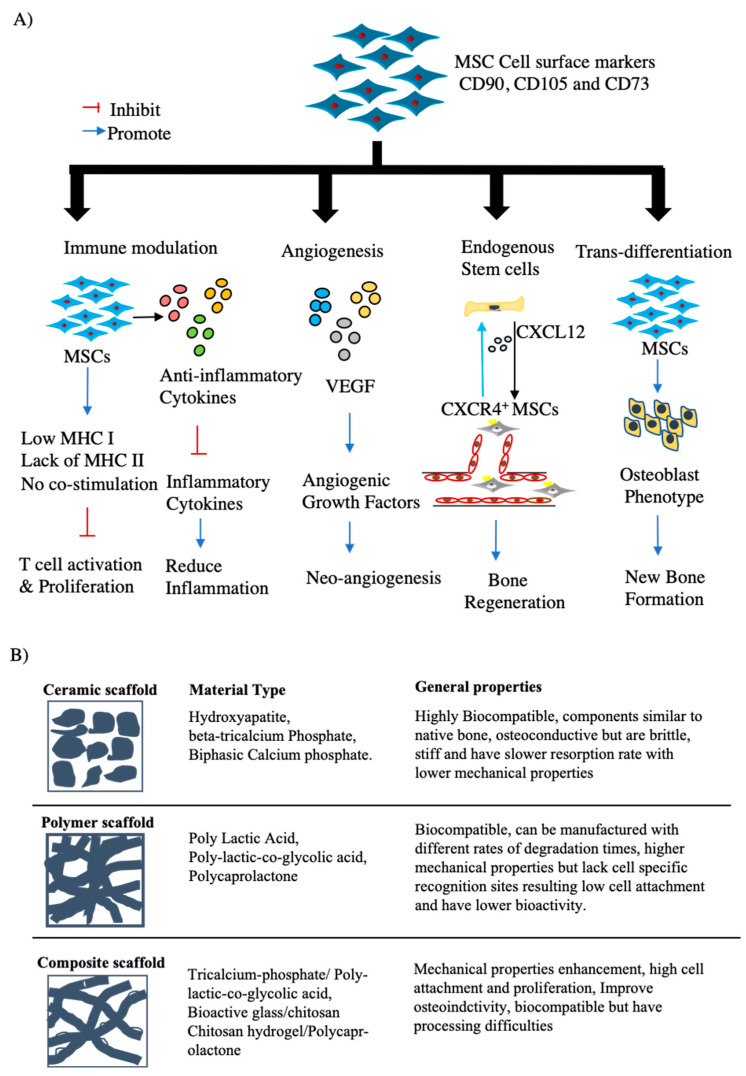
The combination of MSCs and bioscaffold materials used for BTE. (**A**) Mechanisms underlying MSC-based bone regeneration. Due to their characteristic expression of cell markers CD90, CD105, and CD73, and lack of HLA-molecules, MSCs have a bone tissue regeneration capacity through the actions of several mechanisms, including (1) the modulation of immune responses through the prevention of T-cell activation and reduction in the secretion of inflammatory cytokines; (2) the secretion of the angiogenic induction factor VEGF, which helps to form new blood vessels and in turn enhance bone regeneration; (3) the release of chemotactic chemokines at the bone defect site to recruit endogenous stem cells that will further enhance bone regeneration at that location; (4) the trans-differentiation of these cells into osteoblasts under the influence of host-derived factors that helps to promote new bone formation. (**B**) Representation of the routinely used scaffolds with examples and their general properties in the development of BTE technology.

**Figure 3 cells-10-02687-f003:**
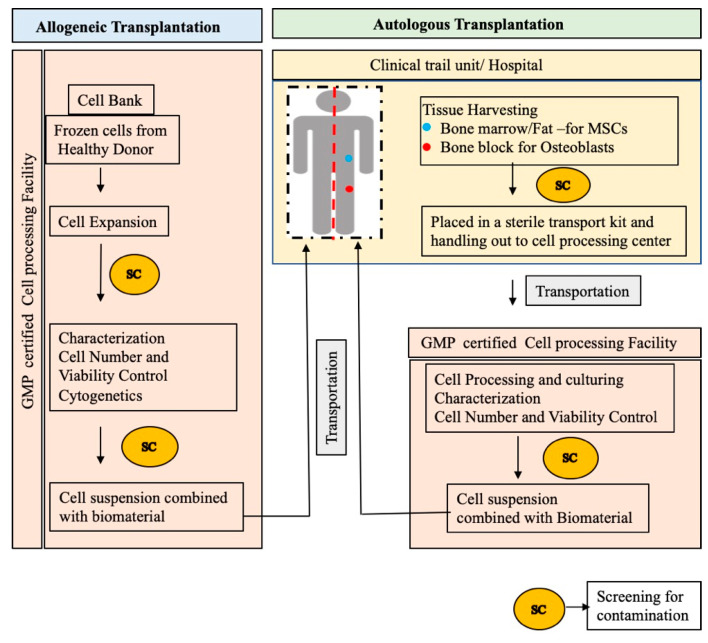
Schematic outline of standard procedure in preparing cell-scaffold constructs for bone regeneration.

**Figure 4 cells-10-02687-f004:**
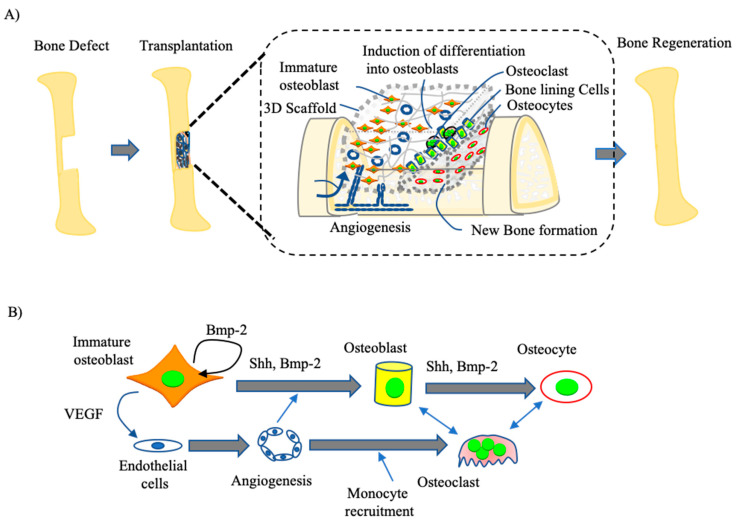
Schematics showing mechanisms of bone regeneration by immature osteoblasts. (**A**) Structures of the osteoblasts seeded scaffold constructs in the bone defect area following transplantation and healing of the defect site. (**B**) Molecular mechanism of bone remodeling by immature osteoblast. The immature osteoblasts under the influence of various cytokines such BMP2, SHH secreted from the bone matrix differentiate into osteoblasts. These osteoblasts produce various cell products, including enzymes alkaline phosphatase and collagenase, growth factors, osteocalcin, and collagen, part of the organic unmineralized component of bone. Few osteoblasts embed inside matrix to become osteocyte and others remain as a bone lining cells on the outer surface. Consequently, when osteoblasts lay down new matrix the osteoclast will differentiate from circulating monocytes/macrophages induced from osteoblasts secreted cytokines such as RANKL and M-CSF, as an inflammatory response to the bone defect from Osteoblasts. Simultaneously, angiogenic factors including VEGF are released from the osteoblasts to form new blood vessels.

**Table 1 cells-10-02687-t001:** Pre-clinical experiments of MSCs-combined with biomaterial for bone regeneration in large animal bone defect models.

Author	Experiment Animal	Type and Size of Defect	Experimental Transplant Groups	Post-Transplant Follow up Period	Outcome
Probst et al., 2020 [50]	Mini pigs	Critical mandibular defect (3 × 1 × 2 cm)	3D TCP-PLGA scaffold seeded with osteogenic differentiated Porcine ADSCs (pADSCs).	12 weeks	pADSCs seeded TCP-PLGA scaffold constructs significantly improved bone regenerations compared to empty scaffold.
Wang et al., 2019 [64]	Rhesus Monkeys	Critical alveolar bone defect (10 × 10 × 5 mm)	3D-Bioactive glass (BG) + BMP/chitosan (CS) + BMMSCs	12 weeks	BMP/CS nanoparticles loaded on 3D-BG scaffold promoted bone regeneration ability in vivo, and preload of BMMSCs promote this ability further.
Hsieh et al., 2019 [65]	domestic Ds-Red pigs	Calvarias defect (8 mm in diameter and 2 mm in depth)	Hemostatic gelatin sponge scaffold seeded with EGFP pig BMMSCs	1, 2, 3 and 4 weeks	Osteoid formation in the scaffolds transplanted with seeded BMMSCs was significantly higher than the control group.
Shi et al., 2019 [66]	Minipigs	Maxillary Intraosseous circular defects (12 mm in diameter and 5 mm in depth)	Bio-Oss/autogenous (Pig Gingival MSCs) pGMSCs (2 × 10^6^)/SB431542 (TGF-β signalling inhibitor).	8 weeks	pGMSCs treated with a TGF- β signaling inhibitor successfully repair minipig severe maxillofacial bone defects.
Qiu et al., 2018 [67]	Minipigs	Lateral femoral condyle defect (8 mm in diameter and 10 mm in depth)	Calcium phosphate cement (CPC) scaffold seeded with autologous BMMSCs plus autologous PRP (CPC-BMSC-PRP, 1 × 10^6^ cells/scaffold)	6 and 12 weeks	CPC scaffold co-delivered BMMSCs-PRP promoted scaffold resorption and doubled bone regeneration in large defects than control groups
Zhang et al., 2017 [68]	Minipigs	Non-healing full thickness cranial defects (2 cm width × 3 cm length × 0.5 cm depth)	IMC (intrafibrillarly-mineralized collagen) scaffold seeded with 1 × 10^6^ PDLSCs cells	12 weeks	Compared with HA, IMC-seeded PDLSCs achieved a significantly higher extent of new bone formation, with the normal architecture of natural bones and blood vessels.
Scarano et al., 2017 [69]	Minipigs	Critical-size circular defects (5 mm diameter; 5 mm thickness) in the mandibular body	Bone porcine block (BPB) scaffold seeded with 100 ul cell suspension of BMMSCs	12 weeks	BPB when used as a scaffold induce bone regeneration and further benefit from the addition of BMMSCs in the tissue-engineered constructs.
Lin et al., 2015 [70]	Minipigs	Massive segmental bone defects (30 mm in length) at the mid-diaphysis of femora	Transduced pig ADSCs loaded onto PLGA scaffold	2, 4, 8 and 12 weeks	ADSCs/scaffold constructs successfully healed massive segmental bone defects at the mid-diaphysis of femora in minipigs significantly than control group.
Cao et al., 2015 [71]	Mini pigs	Calvarial bone defects (3 cm × 1.8 cm oval defect)	BMMSCs pretreated with 75 μg/mL aspirin for 24 h seeded onto hydroxyapatite/tricalcium phosphate (HA/TCP)	6 months	BMMSCs pretreated with aspirin have a greater capacity to repair calvarial bone defects in a mini swine model
Fan et al., 2014 [72]	Rhesus monkeys	Segmental tibial defects (20 mm in length)	Autologous prevascularized BMMSCs (5 × 10^6^)-β-TCP constructs	4, 8 and 12 weeks	Significantly higher amount of neo-vascularization and radiographic grading score in prevascularized BMMSCs-β-TCP constructs

## Data Availability

Data sharing is not applicable to this article.

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
