# Peer review of "Clinical Applications of Cell-Scaffold Constructs for Bone Regeneration Therapy"

_cells, 2021, doi:10.3390/cells10102687_

Round 1

Reviewer 1 Report

The manuscript is a review paper concerning bone tissue engineering. In the introduction, the authors declare a particular emphasis on the available cell sources and consequently, their main conclusion is connected with a particular postulate towards using immature osteoblasts as an alternative to mesenchymal stem cells (MSC), which are currently being widely exploited in the field.

In principle, a fair, careful review of medicinal experiments in the area of bone tissue engineering would be highly expected. As for now, the data is scattered in the scientific literature from various areas. What is more, although bone was chronologically the first tissue of interest in tissue engineering, clinical data on efficacy of bone tissue engineered products are scarce. Therefore the subject is interesting and important.

The advantage of the work is also the fact that the authors formulate their view on the perspective directions, which is an important determinant of a good review.

Unfortunately, the work is written very superficially, it omits important publications, it does not include an in-depth discussion of the quoted data, and the arguments provided do not justify the final thesis.

Detailed comments:

The most crucial issues:

1) Data presented in tables: 1-3 are incomplete.

Just for example:

Stockman and his group observations in pig calvarial defects (DOI: 10.1016/j.jcms.2011.05.004) - should not be omitted in Table 1, Gjerde C. et al (doi.org/10.1186/s13287-018-0951-9) – the same for table 2., etc.

2) For discussion about clinical data, the records from widely used ClinicalTrials database cannot be omitted. There are very many limitations of this source of scientific information, and the data cannot be put under the scientific debate automatically, but – after a careful individual verification of the interesting records - they should be taken into account in a systematic review.

3) A clear/sharp distinction between MSC and preosteoblasts – as applied in table 2 vs table 3 is difficult and at least one example from table 3 may serve as an example. Namely, Mango et al referred in table 3 (table for osteoblasts and not MSC), used the term “osteoblasts” in the Abstract, but if you know the whole text, you know that cells used for observation in this study, were isolated from bone marrow and not subjected to differentiation. Taking into account that the comparison between the observation described in those two tables are the main basis for the conclusions from this review – this is a substantial limitation.

4) Another example of a lack of diligence in presenting the data cited in the work is the second record from the same table (Pradel et al, 2006) – there is a “resorbable bovine collagen matrix” indicated as a scaffold, which may suggest that this is different material as compared to the demineralized bone matrix – used in Pradel et al, 2012 (record 4 from the table). Indeed the authors of the review use the terminology taken from the abstracts. However, there was the same scaffold used in both cases – as described in the full texts.

At the same time, this particular type of scaffold is worth a deeper analysis – the osteogenic properties of demineralized bone tissue are very well known and documented. In both publications mentioned above, the clinical result obtained in a group which have received cells delivered on demineralized bone was  compared to the control group which have received bone transplant. Therefore there is no a good basis to judge about the role of the cells themselves – this issue should at least  be disussed.

Finally, even if the described limitations were to be ignored and if we accept the outcome of the works cited in the text at face value, it provides a weak basis for formulating conclusions proposed in the work.

Minor remarks:

There is no logical rationale for including the section describing scaffold types (2.2.1.) as subsection in Part 2: MSCs for bone regeneration. The same scaffolds are taken into account for bone tissue engineering regardless of the type of cells used.

There are very many places in the  text, which require a reference to specific literature items. For example: page 3, line 85 as one of the most evident. References would be also useful in the tables (actually for Ingo et al. 2006 (table 3) there is no reference at all).

The title of the manuscript is inadequate for 2 reasons – 1) it looks like they are therapies for tissue engineering, while rather the opposite is true (editorial remark), 2) the paper discuss mainly products which are classified in UE, USA, Canada and all the other developed countries as medicinal products, so the term transplantation is unfortunate (formal remark).

Author Response

Thank you very much for constructive suggestions We have undertaken significant revisions, and the manuscript has modified title, a re-arrangement of the sequence of sections, and about 20% of the new text have been added. Moreover, we also provided a new table for clinical trials after reviewing the clinicalTrail.gov database to better understand the type of tissue-engineered constructs used in the present clinical trials. Substantial modification has been done mainly in the topics related to pore size and scaffolds, the role of MSCs in bone regeneration, types of cell-scaffold constructs used in clinical trials with new references, standard procedures for the handling of MSCs, and preparation of MSCs-scaffold constructs before their clinic use. In addition, we also present a new subsection about gene-based therapy for bone regeneration which has potential for future clinical BTE applications. Overall, we believe these modifications present a much clearer context of the goals, scope, and significance of currently available cell-scaffold constructs for clinical use.

Please see the attachmnet for the point-by-point response letter.

Reviewer 2 Report

Comments to the Author(S)

In this review (cells-1360298), the authors discussed the delivery strategies of cells, mainly focusing on MSCs and derived-MSCs for bone tissue repair, followed by concentrating on scaffold-based applications to deliver these cells to the defect side(s). In the end, the authors discussed the pre-clinical and clinical applications of MSCs, and osteoblasts for bone tissue engineering. Although this manuscript has merits, serious shortcomings must be addressed, and significant reorganization and rewriting of the contents are needed.

My recommendation justifications are detailed as follows.

  1. The reviewer thinks that the title of this manuscript is not correctly emphasizing the content of the manuscript. The authors mainly focused on MSCs and derived-MSCs from Adipose, iPSCs, and ESCs for cell therapy. In addition, they discussed the transplantation strategies of these cells relying on 3D constructs. The authors did not provide examples of sole transplantation of cells (scaffold-free) methods as the title indicated "cell transplantation therapy.", which does not mean the delivery of the cells using scaffolds. In addition, there are a variety of different cell types that also plays an essential role in the bone remodeling process, and those cells are not included in this review. Therefore, the authors should consider revising the title.
  2. The authors mentioned critical-sized bone defects (CSDs) frequently in the manuscript; however, they did not clarify what is considered to be CSDs.
  3. The pore structure of scaffolds is crucial as it facilitates bone tissue repair, such as blood vessel ingrowth, which is necessary for cells' survival in the inner area of scaffolds. The authors briefly discussed the importance of the porosity of implanted constructs. Still, they did not clarify the ideal pore size ranges for optimal bone tissue repair and how this parameter could affect the functionality of the delivered cells into the defect site. 
  4. The authors should include a section on gene therapy strategies for bone tissue repair focusing on ex-situ, in-situ, sequential-delivery techniques of cells transfected with plasmid DNAs, growth or differentiation factors, microRNA, etc. What are the advantages or disadvantages of transfecting cells before implantation within the bone defect?
  5. The delivered cells into the defect side can witness challenges, including not enough support of oxygen and nutrients to sustain their viability and support their growth until newer blood vessels are formed. High cell density might prove to be yet another reason for the reduced bone formation. The authors should include a section on how the transplanted cells contribute to bone regeneration despite these challenges. 
  6. Figure 2B is not clear in terms of the representation of scaffold-based bone repair approaches.
  7. What is BCP stand for? Does 'B' mean 'beta'?
  8. The reference in Diloge et al., 2020 in Table 2 does not exist in the manuscript references. 
  9. Similar to Table 3, the authors should include the names of the commercially used materials in provided studies in Table 2.
  10. The authors should discuss cell transplantation methods for combining hybrid strategies (hard and soft biomaterials) for bone tissue repair (i.e., https://doi.org/10.1016/j.msec.2019.110128, https://doi.org/10.1038/s41598-017-13838-7)
  11. The authors should consider adding time points for mentioned pre-clinical studies in Table 1.
  12. The authors mainly discussed cell delivery strategies combined with 3D scaffolds for bone tissue repair. What about scaffold-free methods, solely rely on the implantation of MSCs, bone spheroids, or organoids? The authors should include a section on scaffold-free strategies for cell transplantation for bone repair.
  13. The authors should discuss how MSCs, osteoblasts are handled in the operating room before surgery. What are the rules and regulations that apply to handling and transplantation of MSCs in clinics? What are the acceptable sterilization strategies for MSCs combined with biomaterials before implantation on patients?
  14. The literature in Table 3 is minimal and not recent. The authors should consider expanding the provided information in Table 3.

Author Response

Thank you very much for constructive suggestions We have undertaken significant revisions, and the manuscript has modified title, a re-arrangement of the sequence of sections, and about 20% of the new text have been added. Moreover, we also provided a new table for clinical trials after reviewing the clinicalTrail.gov database to better understand the type of tissue-engineered constructs used in the present clinical trials. Substantial modification has been done mainly in the topics related to pore size and scaffolds, the role of MSCs in bone regeneration, types of cell-scaffold constructs used in clinical trials with new references, standard procedures for the handling of MSCs, and preparation of MSCs-scaffold constructs before their clinic use. In addition, we also present a new subsection about gene-based therapy for bone regeneration which has potential for future clinical BTE applications. Overall, we believe these modifications present a much clearer context of the goals, scope, and significance of currently available cell-scaffold constructs for clinical use.

Please see the attachment for point by point response letter.
